# Extracellular Vesicle-Derived microRNA Crosstalk Between Equine Chondrocytes and Synoviocytes—An In Vitro Approach

**DOI:** 10.3390/ijms26073353

**Published:** 2025-04-03

**Authors:** Catarina I. G. D. Castanheira, James R. Anderson, Emily J. Clarke, Matthias Hackl, Victoria James, Peter D. Clegg, Mandy J. Peffers

**Affiliations:** 1Institute of Life Course and Medical Sciences, University of Liverpool, Liverpool L7 8TX, UKjanders@liverpool.ac.uk (J.R.A.); emily-jayne.clarke2@liverpool.ac.uk (E.J.C.); pclegg@liverpool.ac.uk (P.D.C.); 2Institute of Infection, Veterinary and Ecological Sciences, University of Liverpool, Liverpool L3 5RP, UK; 3TAmiRNA GmbH, 1110 Vienna, Austria; matthias.hackl@tamirna.com; 4School of Veterinary Medicine and Science, University of Nottingham, Nottingham LE12 5RD, UK; victoria.james@nottingham.ac.uk

**Keywords:** 5-ethynyl uridine, equine, extracellular vesicle, microRNA, osteoarthritis, small RNA sequencing, synovial fluid

## Abstract

This study describes a novel technique to analyze the extracellular vesicle (EV)-derived microRNA (miRNA) crosstalk between equine chondrocytes and synoviocytes. Donor cells (chondrocytes, *n* = 8; synoviocytes, *n* = 9) were labelled with 5-ethynyl uridine (5-EU); EVs were isolated from culture media and incubated with recipient cells (chondrocytes [*n* = 5] were incubated with synoviocyte-derived EVs, and synoviocytes [*n* = 4] were incubated with chondrocyte-derived EVs). Total RNA was extracted from recipient cells; the 5-EU-labelled RNA was recovered and sequenced. Differential expression analysis, pathway analysis, and miRNA target prediction were performed. Overall, 198 and 213 miRNAs were identified in recipient synoviocytes and chondrocytes, respectively. The top five most abundant miRNAs were similar for synoviocytes and chondrocytes (eca-miR-21, eca-miR-221, eca-miR-222, eca-miR-100, eca-miR-26a), and appeared to be linked to joint homeostasis. There were nine differentially expressed (*p* < 0.05) miRNAs (eca-miR-27b, eca-miR-23b, eca-miR-31, eca-miR-191a, eca-miR-199a-5p, eca-miR-143, eca-miR-21, eca-miR-181a, and eca-miR-181b) between chondrocytes and synoviocytes, which appeared to be linked to migration of cells, apoptosis, cell viability of connective tissue cell, and inflammation. In conclusion, the reported technique was effective in recovering and characterizing the EV-derived miRNA crosstalk between equine chondrocytes and synoviocytes and allowed for the identification of EV-communicated miRNA patterns potentially related to cell viability, inflammation, and joint homeostasis.

## 1. Introduction

Synovial joints are functional organs constituted by multiple specialized tissues, including cartilage and synovium. Articular cartilage is an avascular connective tissue with unique anisotropic and viscoelastic characteristics that allow for effortless gliding of opposing bone surfaces within the articular joint, facilitating joint movement and load bearing [1,2,3]. Cartilage is solely populated by chondrocytes, which are embedded in an extracellular matrix (ECM) composed of approximately 75% water and a network of type II collagens, proteoglycans and other non-collagenous proteins, inorganic salts, and lipids [1,2,3]. Due to the avascular nature of cartilage, chondrocytes rely on synovial fluid (SF) for functioning and survival [1]. SF is produced by the synovium and acts as a transport medium for nutrients and for numerous regulatory components [4]. The synovium is a vascularized tissue mainly comprising two types of synoviocytes: type A synoviocytes, which are macrophagic cells that possess phagocytic and antigen-presenting capabilities; and type B synoviocytes, which are fibroblast-like cells involved in the production of specialized SF constituents, such as hyaluronan and lubricin, and are the predominant cell type of the synovial intima [5,6]. Communication between chondrocytes and synoviocytes is a dynamic process, and understanding the intricate signaling networks between these two cell types is crucial in the study of joint health and disease. For example, inflammatory cytokine production can be detected in normal synovial tissue because synoviocytes are involved in immunological regulation [7]. However, a marked increase in inflammatory responses can contribute to ECM breakdown and chondrocyte injury, leading to cartilage degradation [8], which highlights the complexity of cellular communication pathways.

The role of extracellular vesicles (EVs) in intercellular communication has gathered significant attention from researchers in recent years [9]. EVs are lipid-bound vesicles secreted by cells into the extracellular space that serve as vehicles for communication [9]. They are released by nearly every cell type and contain lipids, proteins, and genetic material, including a large variety of coding and non-coding types of RNA [10]. The molecular content of EVs can vary greatly from the contents of the respective parental cell, with some studies suggesting that the RNA cargo of EVs skews toward shorter RNA species [10,11]. EV cargo can be specific per vesicle and per cell type and is influenced by the physiological or pathological state of the donor cell [10,11]. For instance, EVs from interleukin-1 beta (IL-1β)-stimulated synovial fibroblasts induced significantly more proteoglycan release from cartilage explants compared with EVs from non-stimulated synovial fibroblasts [12].

There is a growing interest in the role of EV-mediated communication in osteoarthritis (OA) [13]. EVs play important roles in OA progression by aggravating joint inflammation through macrophage activation in SF [14], affecting chondrocyte catabolism and promoting cartilage destruction [15,16], and by regulating subchondral bone remodeling [17]. Therefore, investigating the content of EVs generated by different joint tissues will allow for a better understanding of OA pathogenesis, as well as the discovery of biomarkers for early diagnosis or disease staging [13].

The aim of this study was to develop a novel RNA tracking technique to explore the EV-derived microRNA (miRNA) crosstalk between chondrocytes and synoviocytes and investigate its functional relevance in joint homeostasis. For this, newly synthesized RNA in EV-donor cells was labelled using 5-ethynyl uridine (5-EU); these EVs were then isolated and transferred to EV-recipient cells. Following incorporation by the EV-recipient cells, the labelled RNA was recovered, sequenced and analyzed. The reported technique was effective in recovering and characterizing the EV-derived miRNA. Pathway analysis demonstrated that the most abundant miRNAs appeared to be related to joint homeostasis, while differentially expressed miRNAs appeared to be related to cell viability and inflammation. These data highlight the importance of the EV RNA cargo as a mediator of joint homeostasis and disease.

## 2. Results

### 2.1. Demographics of EV-Donor and EV-Recipient Cells

EV-donor chondrocytes were collected from metacarpophalangeal (MCP) joints of eight horses with a mean (standard deviation [SD]) age of 6.1 (2.6) years, and a mean (SD) joint macroscopic score [14] of 0.8 (0.7). EV-donor synoviocytes were collected from nine horses with a mean (SD) age of 5.8 (2.3) years, and a mean (SD) joint macroscopic score of 1.6 (1.0). Age and joint macroscopic score were similar between EV-donor chondrocytes and synoviocytes (*p =* 0.899 and *p =* 0.092, respectively; Table 1). EV-donor control cells were collected from four additional horses, and the corresponding demographics can be found in the Appendix A.

EV-recipient chondrocytes were collected from MCP joints of five horses with a mean (SD) age of 6.2 (4.4) years and a mean (SD) joint macroscopic score of 2.2 (1.3). EV-recipient synoviocytes were collected from four horses with a mean (SD) age of 5.5 (2.1) years and mean (SD) joint macroscopic score of 2.0 (0.8). Age and joint macroscopic score were similar between EV-recipient chondrocytes and synoviocytes (*p =* 0.810 and *p =* 0.635, respectively; Table 1). EV-recipient control cells were collected from two additional horses, and the corresponding demographics are in the Appendix A.

### 2.2. EV Concentration and Size

A total of 13 sets of 20 mL of cell culture media were collected from EV-donor chondrocytes, with a mean (SD) EV concentration of 1.7 × 10^9^ (1.0 × 10^9^) particles/mL and a vesicle average size (SD) of 186.3 (92.8) nm. Additionally, 14 sets of 20 mL of cell culture media were collected from EV-donor synoviocytes, with a mean (SD) EV concentration of 8.2 × 10^8^ (9.6 × 10^8^) particles/mL and a vesicle average size (SD) of 190.6 (71.7) nm. There were no significant differences in concentration or size between chondrocyte and synoviocyte-derived EVs (*p =* 0.115 and *p =* 0.815, respectively; Table 2).

EVs were pooled together according to EV-donor cell type and equally distributed among EV-recipient cells, so that chondrocytes were incubated with synoviocyte-derived EVs and synoviocytes were incubated with chondrocyte-derived EVs. Overall, EV-recipient chondrocytes were incubated with approximately 4.60 × 10^10^ synoviocyte-derived EVs (3.04 × 10^9^ synoviocyte EVs/mL media) and EV-recipient synoviocytes were incubated with approximately 1.1 × 10^11^ chondrocyte-derived EVs (7.33 × 10^9^ chondrocyte EVs/mL media).

### 2.3. 5-EU RNA Capture from Recipient Cells

After incubating the recipient cells with EVs for 24 h, total RNA was extracted, and the 5-EU-labelled RNA was recaptured and quantified. The mean (SD) concentration of 5-EU-labelled RNA recovered from EV-recipient chondrocytes was significantly lower than of the EV-recipient synoviocytes (43.1 [5.6] ng/μL vs. 62.8 [14.1] ng/μL, respectively; *p =* 0.016; Table 3). The EV-recipient control samples presented a numerically lower mean RNA concentration compared with the experimental samples (Table 3); no statistical analyses were performed due to the sample size of the control groups.

### 2.4. Sequencing Results

#### 2.4.1. Data Overview

The 5-EU-labelled RNA recovered from EV-recipient chondrocytes and synoviocytes was sequenced and analyzed. The total number of reads was similar between EV-recipient chondrocytes and synoviocytes, (5.0 million reads/sample vs. 4.5 million reads/samples, respectively; *p =* 0.905). This was similar to the EV-recipient synoviocyte control sample, with 4.6 million reads. Due to economic constraints, it was not possible to sequence the EV-recipient chondrocyte control sample.

Several types of RNA molecules were identified (Figure 1), including miRNA, transfer RNA, ribosomal RNA, long non-coding RNA, messenger RNA (mRNA), small nuclear RNA, small nucleolar RNA, yRNA, and small cytoplasmatic RNA. The relative abundance of miRNAs varied between 0.10% and 9.62% in EV-recipient chondrocytes and between 0.44% and 7.40% in EV-recipient synoviocytes and had a value of 0.44% in the EV-recipient synoviocyte control. Most samples presented a combined percentage of unclassified and unmapped genomic RNA of over 50% (Figure 1).

A total of 239 miRNAs were identified, of which 213 were present in EV-recipient chondrocytes (RNA originated from EV-donor synoviocytes) and 198 in EV-recipient synoviocytes (RNA originated from EV-donor chondrocytes). Briefly, 43 miRNAs were unique to EV-recipient chondrocytes, 26 miRNAs were unique to EV-recipient synoviocytes, and 170 miRNAs were common to both cell types. Mean (minimum, maximum) absolute miRNA read counts in EV-recipient chondrocytes and synoviocytes were 172,000 (5000–437,000) reads/sample and 167,000 (17,000–431,000) reads/sample, respectively. The top five most abundant miRNAs were the same for both EV-recipient chondrocytes and synoviocytes; these were eca-miR-21, eca-miR-221, eca-miR-222, eca-miR-100, and eca-miR-26a.

##### Predicted mRNA Targets of Unique miRNAs

Ingenuity Pathway Analysis (IPA) software v23.0 (Qiagen, Manchester, UK) was used for target prediction. Using the list of miRNAs that were uniquely identified in EV-recipient chondrocytes (RNA originated from EV-donor synoviocytes), IPA revealed 36 experimentally observed targets for eight miRNAs (Appendix A). The top three diseases and functions associated with the predicted interaction network were “migration of tumor cells” (*p =* 4.36 × 10^−34^; 36/44 molecules involved), “proliferation of connective tissue cells” (*p =* 8.94 × 10^−31^; 28/44 molecules involved), and “invasion of tumor cell lines” (*p =* 2.29 × 10^−30^; 33/44 molecules). For miRNAs that were unique to EV-recipient synoviocytes (RNA originated from EV-donor chondrocytes), target prediction revealed nine experimentally observed targets for three miRNAs (Appendix A). The top three diseases and functions associated with the predicted interaction network were “endometriosis” (*p =* 2.91 × 10^−14^; 9/12 molecules involved), “benign pelvic disease” (*p =* 3.52 × 10^−13^; 10/12 molecules involved), and “pulmonary fibrosis or aplastic anemia” (*p =* 2.77 × 10^−11^; 8/12 molecules involved).

##### Unsupervised Analysis

A heatmap and principal component analysis (PCA) plots were built using reads per million (RPM) normalized and scaled miRNA reads. There was a clear distinction between the miRNA expression of EV-recipient chondrocytes and synoviocytes, which was evident in both the heatmap and the PCA plots (Figure 2).

##### Differential Expression Analysis

miRNAs were filtered to remove low abundant molecules, as later detailed in the methods. In total, 49 miRNAs were analyzed and nine were found to be significantly differentially expressed between EV-recipient chondrocytes and synoviocytes (*p* < 0.05, Table 4). Of these, four were also differentially expressed at a false discovery rate (FDR)-adjusted *p* < 0.05.

##### Target Prediction and Pathway Analysis of Differentially Expressed miRNAs

Target prediction revealed 23 experimentally observed targets for seven out of the nine differentially expressed miRNAs (Appendix A). IPA “Core Analysis” of the combined list (interactome) of differentially expressed miRNAs and miRNA targets revealed that the two top canonical pathways associated with these molecules were “role of osteoblasts in rheumatoid arthritis signaling pathway” (*p =* 2.30 × 10^−10^; overlap 3.5% [8/228]) and “osteoarthritis pathway” (*p =* 2.64 × 10^−10^; overlap 3.4% [8/232]; Appendix A). When analyzing diseases and functions related to these molecules, 26 were related to organismal injury and abnormalities and 23 were related to inflammatory disease (Appendix A). Pathway analysis also revealed significant links to migration of cells (*p =* 3.53 × 10^−19^), angiogenesis (*p =* 7.79 × 10^−17^), apoptosis (*p =* 1.11 × 10^−14^), cell viability of connective tissue cells (*p =* 1.28 × 10^−15^), inflammation of joints (*p =* 3.92 × 10^−13^), and rheumatoid arthritis (*p =* 9.27 × 10^−12^), among others (Figure 3).

## 3. Discussion

This study reported a novel method of tracking EV-contained miRNA cargo between joint cells in vitro, specifically the crosstalk between equine chondrocytes and synoviocytes. Using a 5-EU-labelling and affinity recovery technique, the present study tracked the EV-led delivery of miRNAs from chondrocytes to synoviocytes and vice versa. Subsequent small RNA sequencing analyses revealed that the most abundant miRNA molecules recovered from EV-recipient cells may play important roles in joint homeostasis, and that the content of EVs produced by chondrocytes differs from that of synoviocytes.

Click chemistry has been widely used as a fast and highly selective method of molecular labelling that allows for the formation of covalent links between unnatural chemical groups or amino acids to any protein sites of interest [19]. A previous study investigating the communication of prostate cancer cells and osteoblasts via EV RNA reported the use of a 5-EU-based click chemistry technique to track the transference of miRNAs from donor to recipient cells [20]. The present study used a similar technique, which comprised four main steps: firstly, EV-donor cells were incubated with 5-EU, a non-toxic, alkyne-modified analogue of uridine that is naturally incorporated into newly synthesized RNA; this step enabled the incorporation of the 5-EU label in newly formed RNA in EVs. Secondly, EVs were isolated from donor cells and incubated with recipient cells, allowing for the transference of the 5-EU-labelled RNA within the EVs. Thirdly, total RNA was isolated from the recipient cells and the 5-EU-labelled RNA was biotinylated; during this process, the alkyne-modified uridine of the 5-EU label was combined with an azide-modified biotin through a copper catalyzed click reaction creating a biotin-based handle. Lastly, the total RNA was mixed with streptavidin magnetic beads, and the 5-EU-labelled RNA was captured by the beads through the previously created biotin-based handle [21]. RNA concentrations from experimental samples were compared with unlabeled controls to confirm the success of the 5-EU-labelled RNA transfer. The mean RNA concentration of EV-recipient chondrocytes was numerically lower than that of the corresponding controls (43.1 ng/μL [*n* = 5] vs. 0.9 ng/μL [*n* = 1], respectively). Similarly, mean RNA concentration of EV-recipient synoviocytes was numerically lower than that of the corresponding controls (62.8 ng/μL [*n* = 4] vs. 1.8 ng/μL [*n* = 1], respectively). While statistical analyses between experimental and control groups were not possible due to the small sample size of the control groups, there was a clear difference between the mean values, and therefore the technique was deemed successful. This was further supported by the unsupervised analysis of miRNA sequencing data, which showed a clear distinction between the EV-recipient synoviocyte group and the EV-recipient control synoviocyte sample.

This study quantified EVs using nanoparticle tracking analysis and distributed them evenly across recipient cells, ensuring that flasks of recipient cells were incubated with equal volumes of EV-containing media. This resulted in EV-recipient chondrocytes being incubated with approximately 3.04 × 10^9^ synoviocyte EVs/mL media and EV-recipient synoviocytes being incubated with 7.33 × 10^9^ chondrocyte EVs/mL media. Previous studies have shown that the incorporation of EV cargo may depend on the type of recipient cells [22], and it is still unclear what the ratio of EV/target cell should be to mimic physiological conditions [23]. In some of these studies, the amount of EVs used was calculated based on their protein content and reported as the ratio of EV protein (μg) to the number of recipient cells (e.g., 0.5–50 μg/1 × 10^4^ cells) [22].

Chondrocytes have prolonged metabolic rates, which are partially explained by the lack of vascularization in cartilage and consequent deficient access to nutrients and oxygen [24,25]. There is a known relationship between the capillarity and metabolic requirements of a tissue [26]. With the synovium being highly vascularized [6], it can be assumed that the metabolic activity of synoviocytes is higher than that of chondrocytes. We hypothesized that this could have an impact on the number of EVs isolated from both cell types, as well as the volume of their cargo. This study found that that the number and size of EVs produced by chondrocytes was similar to those produced by synoviocytes. The concentration of 5-EU-labelled RNA recovered from recipient synoviocytes (corresponding to RNA derived from chondrocyte EVs; mean [SD] 62.8 [14.1] ng/μL) was significantly higher than that recovered from chondrocytes (corresponding to RNA derived from synoviocyte EVs; mean [SD] 43.1 [5.6] ng/μL; *p =* 0.016). This indicated that, on average, there was a higher concentration of labelled RNA transferred from chondrocyte-derived EVs to synoviocytes compared with the amount transferred from synoviocyte-derived EVs to chondrocytes. However, the results showed a higher number of miRNAs in synoviocyte-derived EVs compared with chondrocyte-derived EVs, as well as a higher number of miRNAs uniquely identified in synoviocyte-derived samples compared with chondrocyte-derived samples. These findings suggest that a difference in metabolic rate alone might not be a determining factor for the volume and type of EV-led communication between chondrocytes and synoviocytes. Of note, the cell type, cell density, growth medium, and EU concentration might influence the label uptake [21], and future studies may benefit from optimizing assay conditions.

Using small RNA sequencing, this study found a wide variety of small RNA molecules in chondrocyte and synoviocyte-derived EVs, which is in accordance with previous studies [27,28]. Of note, most samples in this study presented a combined unclassified and unmapped genomic percentage of over 50%. “Unclassified” indicates reads that were mapped against the genome but were not found in any of the RNA specific databases, while “unmapped” refers to reads that could not be found in the given reference genome. Even with improvements in computational assembly techniques that have contributed to the refinement of the reference genome for the domestic horse [29], annotation of the equine genome still falls short of the level observed in the human genome, and it is reasonable to expect a higher percentage of unmapped reads. Additionally, the samples used in this study had low RNA content and relatively low complexity. Considering that a low amount of starting material can lead to increased technical noise [30], this might have led to an increased presence of unclassified reads in our sequencing data.

The five most abundant miRNAs were the same for chondrocyte- and synoviocyte-derived EVs and included eca-miR-21, eca-miR-221, eca-miR-222, eca-miR-100, and eca-miR-26a. Previous studies have demonstrated that miR-21, miR-221, and miR-222 are mechanically regulated miRNAs that affect joint homeostasis by regulating load-induced cartilage behavior in healthy and diseased joints [31]. A study investigating the mechano-regulation of miRNAs in bovine articular cartilage reported a significant increase in the expression levels of miR-21, miR-221, and miR-222 in ex vivo cartilage explants that were subjected to increasing load magnitude, as well as in in vivo cartilage exposed to abnormal loading [31]. Similarly, a miRNA microarray analysis of bovine articular cartilage found miR-221 and miR-222 to be upregulated in the anterior weight-bearing articular area of femoral cartilage compared with the posterior non-weight-bearing area in healthy juvenile animals [32]. Additionally, EV-mediated transfer of miR-221-3p between chondrocytes and osteoblasts was shown to have a mechanosensitive function inside cartilage in physiological conditions in an in vitro mice model [33]. It was theorized that, when affected by mechanical loading, chondrocytes secrete EVs that contain miR-221-3p, which can be taken up via the microchannel network of the subchondral bone and inhibit osteoblastic function [33]. Studies investigating the expression and function of miRNAs in OA reported that miR-222 was downregulated in OA chondrocytes [34], and that downregulation of miR-221-3p contributed to cartilage degradation in IL-1β-treated chondrocytes [35], further confirming the regulatory functions of these miRNAs in joints. Together with these findings, our results suggest that EV-led distribution of eca-miR-21, eca-miR-221, and eca-miR-222 may play an important role in maintaining homeostasis in healthy equine joints by conferring the necessary mechanisms for appropriate biomechanical responses to joint loading. The results of the present study also suggest that both chondrocytes and synoviocytes contribute to the regulation of these mechanisms. Notably, miR-21 has been identified in SF EVs in a post-traumatic equine model of OA and was upregulated in OA SF at 10 days post-induction compared with baseline [36].

A study investigating the role of infrapatellar fat pad mesenchymal stem cell-derived exosomes in OA found that miR-100-5p-abundant exosomes protected articular cartilage and ameliorated gait abnormalities [37]. Additionally, overexpression of miR-26a by intraarticular injection significantly attenuated OA progression in mice [38]. In the present study, miR-100 and miR-26a were highly abundant in chondrocyte- and synoviocyte-derived EVs, suggesting that EV-led communication of miR-100 and miR-26a may be implicated in chondroprotective functions in healthy equine joints.

Despite some commonalities, unsupervised analysis of miRNA sequencing data showed a clear separation between chondrocyte- and synoviocyte-derived EV content, demonstrating that these samples can be differentiated based on miRNA expression profiles. Results showed 43 miRNAs that were specific to synoviocyte-derived EVs. Using the miRNA target prediction function in IPA v23.0, this study found 36 experimentally observed targets that were significantly linked to the proliferation of connective tissues as well as the development of vasculature and angiogenesis. Angiogenesis is a fundamental process for growth and for tissue repair after injury [39]. This dynamic process is finely tuned by the interplay of proangiogenic and antiangiogenic factors, and a disruption of such mechanisms can lead to pathological neovascularization of tissues [40].

While the synovium is highly vascular, vascular proliferation is not a feature of the normal joint and is generally associated with inflammation [39]. In fact, angiogenesis and inflammation are closely integrated processes that contribute to the development of OA [40]. Therefore, regulation of angiogenesis helps maintain appropriate tissue structure and function [39]. The transfer of EVs containing miRNAs that target proteins involved in angiogenesis could help regulate angiogenic processes, allowing the joint to respond to microtrauma while preventing OA development or other joint alterations.

The present study found that EVs derived from synoviocytes had an increased expression of miRNAs that regulate angiogenesis and other proliferative processes compared with EVs derived from chondrocytes. Because chondrocytes can induce vascular invasion of the subchondral bone through the production of angiogenic factors such as vascular endothelial growth factor (VEGF) [41], it is possible that EV-led communication of these specific miRNAs from synoviocytes to chondrocytes is an important mechanism for regulating angiogenic homeostasis in the joint. The sources of angiogenic signals to the subchondral bone are not yet understood [42], and it is plausible that synoviocyte-derived EVs may participate in this process.

The present study found 26 miRNAs that were specific to chondrocyte-derived EVs. IPA indicated that there were nine experimentally observed targets, including insulin receptor substrate 1 (IRS1), vascular cell adhesion molecule 1 (VCAM1), and VEGFA. Analysis of the top 10 predicted functions and diseases associated with the target mRNAs revealed processes somewhat irrelevant to joint homeostasis, such as endometriosis and benign pelvic disease. This was probably due to the small list of miRNAs used as input, as well as the stringent filters applied. Still, all experimentally observed targets had been previously shown to influence joint homeostasis to some degree. For example, IRS1 is important for bone turnover [43], while VCAM is important in leukocyte trafficking in the synovium [44]. VEGFA is a chondrocyte survival factor during development, and it is crucial for bone formation and skeletal growth in early postnatal life [45]. Additionally, VEGFA is increased in OA joints and synovial levels of this molecule significantly correlate with clinical manifestations, functional impact, and radiological changes in knee OA [46]. These examples support the hypothesis that the differentially expressed miRNAs found in this study are involved in processes of joint homeostasis and/or disease.

Equine models are often used in translational studies of OA to aid the investigation of disease pathways and potential therapeutic candidates due to similarities in disease pathogenesis, clinical presentation, and pathological changes between horses and humans [47,48]. In the field of EVs specifically, current literature suggests that horses provide a suitable model to investigate the diagnostic and therapeutic potential of EVs in OA [49,50]. The present study was the first to report a technique that allows for the investigation of the EV-led miRNA crosstalk between chondrocytes and synoviocytes in equine cells, and no studies of a similar nature have been reported in humans. While further research is needed to ascertain the reproducibility of this technique in human joint cells, the results of this study support the hypothesis that EV-RNA cargo is a mediator of joint homeostasis and disease.

This study had some limitations. Owing to constraints in acquiring fresh equine tissues and technical difficulties in culturing primary cells, sample availability was limited, which led to a relatively small sample size and may have affected the statistical power of the analyses. Additionally, the presence of EV surface markers was not investigated, and morphological evaluation was not performed, meaning that EVs were not fully characterized; however, the current unavailability of validated equine antibodies represents a significant obstacle in the characterization of equine EVs [51]. While EV-led RNA transfer was successful in this study, the uptake of EV cargo does not necessarily equate to functionality. Future studies may benefit from including functional tests to confirm the active incorporation of transferred molecules into recipient cells—for example, by assessing the expression of target genes or proteins by reverse transcription quantitative polymerase chain reaction or Western blot.

## 4. Materials and Methods

All materials and reagents used in this experiment were from ThermoFisher Scientific, Paisley, UK, unless otherwise specified. An overview of the experimental protocol is shown in Figure 4.

### 4.1. Tissue Collection

Equine distal limbs were obtained from the abattoir as a by-product of the agricultural industry. The Animal (Scientific procedures) Act 1986, Schedule 2, does not define collection from these sources as scientific procedures, and ethical approval was therefore not required. The MCP joints were aseptically dissected, photographed, and scored by three independent researchers using the Osteoarthritis Research Society International equine macroscopic grading system [14]. The final joint macroscopic score was an average of the scores obtained by the three researchers. Horses were selected for inclusion based on a joint macroscopic score of <3 and an age of <15 years old.

### 4.2. Isolation of Primary Chondrocytes and Synoviocytes

Equine chondrocytes and synoviocytes were isolated as previously described [52,53]. Briefly, the MCP joints were aseptically dissected, and multiple pieces of cartilage (from the metacarpus, proximal phalanx, and sesamoids) and synovium were collected. Tissues were transferred to a class II laminar flow hood, cut into 1–3 mm pieces, and digested overnight using a collagenase type II solution (Worthington Biochemicals, Lakewood, NJ, USA). Once strained and washed, the cells were cultured in monolayer at a density of 20,000 cells/cm^2^ with complete media (Dulbecco’s Modified Eagle Medium [DMEM] supplemented with 10% fetal bovine serum [FBS], 1% penicillin–streptomycin [Pen/Strep], and 0.2% amphotericin B), and incubated at 37 °C. Chondrocytes were incubated in hypoxic conditions (5% O_2_ and 5% CO_2_) and synoviocytes were incubated in normoxic conditions (25% O_2_ and 5% CO_2_). Complete media changes were carried out every two days until the cells reached 80% confluence. The cells were stored in liquid nitrogen until further use.

### 4.3. EV-Donor Cell Set Up and RNA Labelling

EV-donor cells were thawed, cultured in T175 flasks with complete media at a density of 20,000 cells/cm^2^, and used at passage 1 (P1). Complete media changes were performed every two days until the cells reached 50–60% confluency. The flasks were then washed thrice with sterile phosphate buffered saline (PBS) and incubated with 20 mL of EV media (phenol red-free DMEM supplemented with 1% Pen/Strep, 0.2% amphotericin B, and 1% L-glutamate), with the addition of 5-EU at 1 μL/mL. Four controls (chondrocyte, *n* = 2; synoviocyte, *n* = 2) were cultured in parallel in EV media with no added label. Cells were incubated for 48 h, after which the cell culture media was collected from each flask and stored in the fridge until further processing. The 5-EU treatment was repeated, and a second batch of media was collected after 48 h.

### 4.4. EV Isolation and Characterization

Culture media batches were pre-processed using sequential centrifugation (300× *g* for 10 min, 2000× *g* for 10 min, and 10,000× *g* for 30 min). The resulting supernatant was concentrated down to approximately 400 μL using a 20 mL Vivaspin column (Sartorius, Göttingen, Germany). Samples were transferred to a class II laminar flow hood, and EVs were isolated through size exclusion chromatography using an IZON qEV concentration kit with qEV single columns (IZON, Lyon, France). Firstly, the qEV single columns were flushed with filtered sterile PBS. Then, 150 μL of sample was added to the column, followed by 1 mL of filtered PBS. The first five 200 μL fractions that flowed through the column were discarded, and an additional 1 mL of filtered PBS was added. The following five 200 μL fractions (containing EVs) were collected.

For size analysis and quantitation, 100 μL were aliquoted from each EV sample and subsequently diluted 1:10 in filtered PBS. Diluted samples were characterized using the NanoSight NS300 (Malvern Panalytical, Great Malvern, UK), which was run at 25 °C, with the camera level, slider shutter, and slider gain set at 10, 696, and 73, respectively. Three 60 s-long videos were recorded for each sample, and the results were averaged. For image analysis, the detection threshold was set at 5. Data analysis was performed using NTA 3.4 software (Malvern Panalytical, Malvern, UK).

### 4.5. EV-Recipient Cell Setup and EV Incubation

EV-recipient cells were thawed, cultured in T75 flasks with complete media at a density of 20,000 cells/cm^2^, and used at P2. Complete media changes were performed every two days until the cells reached 40–50% confluency. Once confluent, the flasks were washed thrice with sterile PBS. Chondrocyte-derived EVs were pooled together and distributed among the EV-recipient synoviocyte flasks in equal volumes. Similarly, synoviocyte-derived EVs were pooled together and evenly split among EV-recipient chondrocyte flasks. Finally, control recipient chondrocytes (*n* = 1) were incubated with EVs derived from control donor synoviocytes, and control recipient synoviocytes (*n* = 1) were incubated with EVs from control donor chondrocytes. All cells were incubated with EVs for 24 h. After this period, cells were trypsinized and kept in TRIzol until further processing.

### 4.6. RNA Extraction

Total RNA was extracted from EV-recipient cells using the miRNeasy^®^ Serum/Plasma Advanced kit (Qiagen, Manchester, UK) following the manufacturer’s protocol, with the recommended adaptations according to an input volume of 500 μL. RNA was eluted in 18 μL of RNAse-free water.

### 4.7. Biotinylation of RNA and RNA Capture

5-EU-labelled RNA was recovered from total RNA samples using the Click-iT^®^ Nascent RNA Capture Kit (ThermoFisher Scientific, Paisley, UK), following the manufacturer’s protocol with small adjustments. Briefly, a master mix containing Click-iT^®^ reaction buffers, copper sulphate (component D), and biotin azide (component C) was incubated with total RNA to form a click reaction. To precipitate the RNA, a solution containing ultrapure glycogen, ammonium acetate, and chilled absolute ethanol was added to the click reaction. The samples were incubated at −80 °C overnight and centrifuged at 13,000× *g* for 20 min at 4 °C. After discarding the supernatant, 75% ethanol was added to the RNA pellet, and the samples were centrifuged again at 13,000× *g* for 5 min. The ethanol was removed and the pellets were allowed to air dry.

The pellet was resuspended in 20 μL of Ultra-Pure™ DNase/RNase-free distilled water and incubated with the kit’s RNA-binding reaction mix at 69 °C for 5 min. A total of 25 μL of Dynabeads^®^ MyOne™ Streptavidin T1 (ThermoFisher Scientific, Paisley, UK) was added to the RNA solution and incubated for an additional 30 min at room temperature. Finally, the beads were immobilized using a magnetic stand and washed multiple times with the kit’s reaction buffers. The beads were resuspended in 25 μL of Reaction Wash Buffer 2. One μL of the resuspended bead solution was used for RNA quantification using the NanoDrop 8000 (ThermoFisher Scientific, Paisley, UK).

To precipitate the RNA from the beads, the samples were resuspended in 10 mM tris-hydrochloride and incubated at 70 °C for 5 min. The beads were separated from the solution using a magnetic rack. The precipitation step described above was repeated. The dry pellet was resuspended in 35 μL of RNAse-free distilled water.

### 4.8. Small RNA Sequencing

In brief, 8.5 μL of RNA was used as an input for small RNA sequencing library preparation. One μL of miND^®^ spike-in standards (TAmiRNA GmbH, Vienna, Austria) was added to each sample prior to small RNA library preparation using the RealSeq Biofluids library preparation kit (RealSeq Biosciences, Santa Cruz, CA, USA). Adapter-ligated libraries were amplified with 25–28 PCR cycles using barcoded Illumina reverse primers in combination with the Illumina forward primer (Illumina, San Diego, CA, USA). Library quality control was performed using DNA1000 Chip (Agilent, Santa Clara, CA, USA). An equimolar pool consisting of all sequencing libraries was prepared and sequenced on an Illumina Novaseq (Illumina, CA, USA) with 100 bp single end reads.

### 4.9. Data Analysis

The horses’ demographics, EV characteristics, and RNA concentration were analyzed descriptively and compared between groups (chondrocytes vs. synoviocytes). Data were tested for normality of distribution and statistical tests were selected for the specific data types, as appropriate; all tests were performed using GraphPad Prism version 8.0 for Windows, with significance set at a *p*-value of 0.05.

#### 4.9.1. Small RNA Sequencing Analysis

Data were processed using the miND pipeline and adapted to equine samples [54,55]. Briefly, the overall quality of the next-generation sequencing data was evaluated automatically and manually with fastQC v0.11.9 [56] and multiQC v1.10 [57]. Reads from all quality samples were adapter-trimmed and quality-filtered using cutadapt v3.3 [58] and filtered for a minimum length of 17 nucleotides. Mapping steps were performed with Bowtie v1.3.0 [59] and miRDeep2 v2.0.1.2 [60]. Reads were mapped against the genomic reference EquCab.3.0 provided by Ensembl (Cambridge, UK) [61], allowing for two mismatches, and subsequently against miRBase v22.1 [62], then filtered for miRNAs of eca only, allowing for one mismatch. For a general RNA composition overview, non-miRNA mapped reads were mapped against RNAcentral [63] and then assigned to various RNA species of interest. Statistical analysis of pre-processed sequencing data was undertaken with R v4.0 and the packages pheatmap v1.0.12, pcaMethods v1.82, and genefilter v1.72 to generate heatmaps. Differential expression analysis with edgeR v3.32 [64] used the quasi-likelihood negative binomial generalized log-linear model functions provided by the package. The independent filtering method of DESeq2 [65] was adapted for use with edgeR to remove low abundant miRNAs (defined as RPM values that are lower than 10 divided by the smallest library size in at least half the number of samples of the smaller group) and thus optimize FDR correction.

#### 4.9.2. Target Prediction and Pathway Analysis

Prediction of functional targets for the groups of miRNAs determined by small RNA sequencing analysis to be (1) uniquely expressed in chondrocytes, (2) uniquely expressed in synoviocytes, and (3) differentially expressed (*p* < 0.05) between chondrocytes and synoviocytes was carried out using IPA v23.0 (Qiagen, Manchester, UK). Each equine miRNA was matched to its human equivalent using miRbase, and the corresponding miRbase IDs were input into IPA as identifiers. Of note, there was no filtering step prior to inputting miRNA data into IPA. Data were then analyzed using the “Target Prediction” function in IPA, and the results were filtered for experimentally observed targets in the tissue type “cartilage” and the cell types “chondrocytes” and “osteoblasts”. The interactions between miRNAs and their predicted mRNA targets were displayed as networks using the “Path Designer” tool. Additionally, the “Grow—Diseases & Functions” function was used to assess the top 10 predicted diseases and functions associated with each of the predicted networks.

Once mRNA target prediction was completed, the list of differentially expressed miRNAs (along with their expression values [log fold change] and significance levels obtained by small RNA sequencing) was combined with the list of predicted mRNAs and input back into IPA. Data were analyzed using the “Core Analysis” function, which calculates the *p*-value of overlap between the molecules in the dataset with the disease and functions contained in the Ingenuity Knowledge Base using a right-tailed Fisher’s exact test. This analysis also uses an algorithm to generate a network of canonical pathways, biological functions, diseases, and network-eligible molecules based on their connectivity. The results were filtered for experimentally observed associations only. The interaction networks were visualized using the “Path Designer” function.

## 5. Conclusions

Understanding the intricate crosstalk between chondrocytes and synoviocytes is crucial to better comprehend the mechanisms underlying joint health and disease. This proof-of-concept study reported a novel method of tracking EV-contained miRNA cargo between joint cells—specifically, the crosstalk between equine chondrocytes and synoviocytes. Through a sequencing analysis of the EV content, a list of highly abundant miRNAs in EVs that were common for chondrocytes and synoviocytes and appeared to be related to joint homeostasis were found. There were nine differentially expressed miRNAs that appear to be related to cell viability and inflammation. While further studies are needed to confirm the active incorporation of these EV-transferred miRNAs and explore the subsequent biological response, the results of this study support the hypothesis that EV-RNA cargo is a mediator of joint homeostasis and disease.

## Figures and Tables

**Figure 1 ijms-26-03353-f001:**
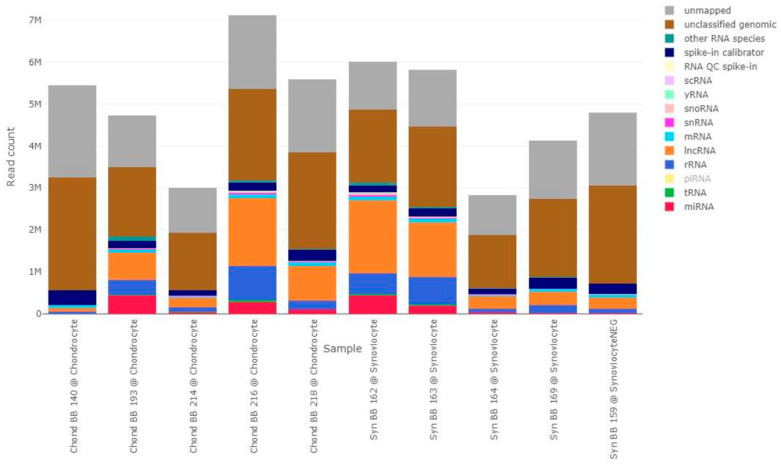
Absolute read composition of all samples, stratified by RNA type. Different colors represent different RNA types, as indicated in the figure label. Labels with lower opacity correspond to RNA types that were not identified in this dataset (piRNA). The bars represent the different samples, with the sample ID and group indicated below each bar. “Chond” refers to 5-EU-labelled RNA isolated from EV-recipient chondrocytes. “Syn” refers to 5-EU-labelled RNA isolated from EV-recipient synoviocytes. lncRNA, long non-coding RNA; miRNA, microRNA; mRNA, messenger RNA; piRNA, piwi interfering RNA; QC, quality control; rRNA, ribosomal RNA; scRNA, small conditional RNA; snRNA, small nuclear RNA; snoRNA, small nucleolar RNA; tRNA, transfer RNA.2.4.2. EV-Transferred miRNAs.

**Figure 2 ijms-26-03353-f002:**
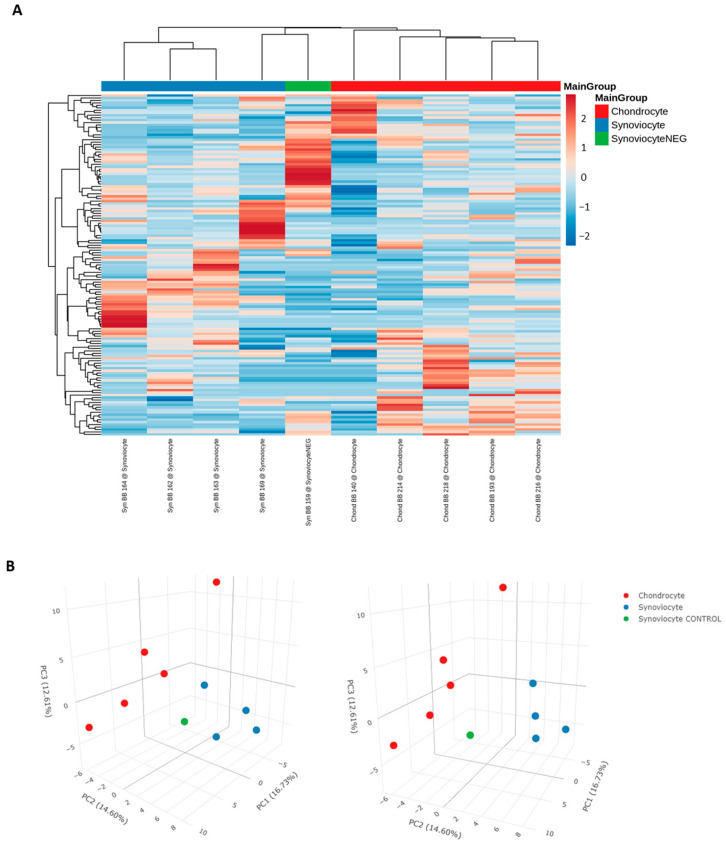
(**A**) Heatmap and (**B**) PCA plot of normalized and scaled miRNA reads. Scaling of data for the heatmap was undertaken using the unit variance method for visualization in heatmaps and filtered to show a minimum of 5 RPM in at least 50% of samples. Each column represents a sample, each row represents a miRNA, and the color intensity of each cell corresponds to the miRNA expression in a given sample. For the PCA plot, reads were normalized as RPM and scaled using unit variance, and principal components were calculated using singular value decomposition with imputation. Colors represent the different sample types, as shown in the figure label. “Synoviocyte” refers to labelled RNA isolated from EV-recipient synoviocytes. “Chondrocyte” refers to labelled RNA isolated from EV-recipient chondrocytes. “SynoviocyteNEG” and “Synoviocyte CONTROL” refer to the EV-recipient control synoviocyte sample.

**Figure 3 ijms-26-03353-f003:**
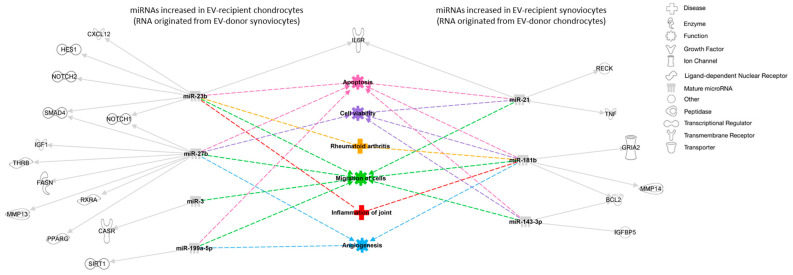
Interaction network of miRNA targets, diseases, and functions predicted to be associated with the differentially expressed miRNAs between EV-recipient chondrocytes and synoviocytes. miRNAs are represented in gray, experimentally predicted targets are represented in white, and predicted cellular functions and diseases are represented in different colors. Solid arrows represent direct relationships, and dashed arrows represent predicted diseases and functions associated with the different molecules. BCL2, B-cell lymphoma 2 apoptosis regulator; CASR, calcium-sensing receptor; CXCL12, C-X-C motif chemokine ligand 12; FASN, fatty acid synthase; GRIA2, glutamate ionotropic receptor AMPA type subunit 2; HES1, hes family bHLH transcription factor 1; IGF1, insulin-like growth factor 1; IGFBP5, insulin-like growth factor-binding protein 5; IL6R, interleukin 6 receptor; MMP, matrix metalloproteinase; NOTCH, notch receptor; PPARG, peroxisome proliferator-activated receptor gamma; RECK, reversion-inducing cysteine-rich protein with Kazal motifs; RXRA, retinoid X receptor alpha; SIRT1, sirtuin 1; SMAD4, mothers against decapentaplegic homolog 4; THRB, thyroid hormone receptor beta; TNF, tumor necrosis factor.

**Figure 4 ijms-26-03353-f004:**
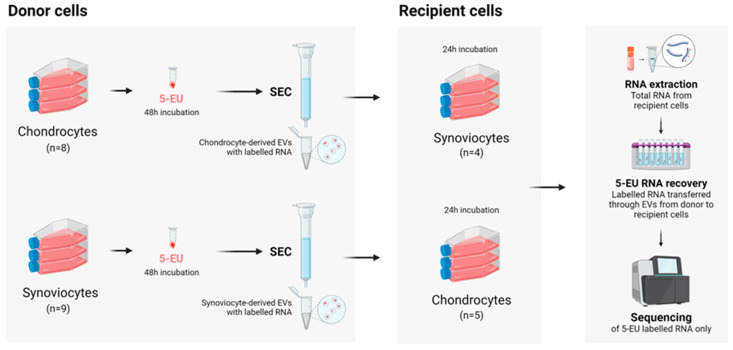
Overview of the experimental protocol. Figure created with BioRender.com (www.biorender.com [accessed on 1 January 2024]). SEC, size exclusion chromatography.

**Table 1 ijms-26-03353-t001:** Demographics of EV-donor and -recipient cells, stratified by cell type.

Characteristics	EV-Donor Cells		EV-Recipient Cells	
	Chondrocytes (*N* = 8)	Synoviocytes (*N* = 9)	Chondrocytes (*N* = 5)	Synoviocytes (*N* = 4)
Age, years				
Mean (SD)	6.1 (2.6)	5.8 (2.3)	6.2 (4.4)	5.5 (2.1)
95% CI	4.0–8.3	4.0–7.5	0.7–11.7	2.2–8.8
Min; Max	3; 10	3; 10	3; 14	3; 8
*p*-value	0.899 ^1^		0.810 ^1^	
Sex, *n* (%)				
Female	3 (60.0) ^2^	5 (62.5) ^3^	0 (0.0) ^4^	3 (75.0)
Male	2 (40.0) ^2^	3 (37.5) ^3^	1 (100.0) ^4^	1 (25.0)
*p*-value	0.928 ^5^		–	
Joint macroscopic score (0–9) ^6^, *n* (%)				
Scores				
0	3 (37.5)	2 (22.2)	1 (20.0)	0 (0.0)
1	4 (50.0)	1 (11.1)	0 (0.0)	1 (25.0)
2	1 (12.5)	5 (55.6)	1 (20.0)	2 (50.0)
3	0 (0.0)	1 (11.1)	3 (60.0)	1 (25.0)
≥4	0 (0.0)	0 (0.0)	0 (0.0)	0 (0.0)
Mean (SD)	0.8 (0.7)	1.6 (1.0)	2.2 (1.3)	2.0 (0.8)
95% CI	0.2–1.3	0.8–2.3	0.6–3.8	0.7–3.3
Min; Max	0; 2	0; 3	0; 3	1; 3
*p*-value	0.092 ^7^		0.635 ^1^	

^1^ Calculated using a Mann–Whitney U test. ^2^ *n1* = 5. ^3^ *n1* = 8. ^4^ *n1* = 1. ^5^ Calculated using a Chi-squared test. ^6^ Average of the scores obtained by three independent researchers using the Osteoarthritis Research Society International scoring system [18]. ^7^ Calculated using an unpaired *t*-test.CI confidence interval; Max, maximum; Min, minimum; n, number of horses included in a given category; N, number of horses included in the analysis set; N1, number of horses with available information in a given category; SD, standard deviation.

**Table 2 ijms-26-03353-t002:** Size analysis and quantitation of EVs, stratified by EV-donor cell type.

EV Characteristics	Chondrocytes (*N* = 8)	Synoviocytes (*N* = 9)
EV concentration, particles/mL		
Mean (SD)	1.7 × 10^9^ (1.0 × 10^9^)	8.2 × 10^8^ (9.6 × 10^8^)
95% CI	8.5 × 10^8^–1.5 × 10^9^	5.3 × 10^8^–1.1 × 10^9^
Min; Max	7.7 × 10^7^; 3.0 × 10^9^	7.9 × 10^7^; 2.6 × 10^9^
*p*-value	0.115 ^1^	
EV size, nm		
Mean (SD)	186.3 (92.8)	190.6 (71.7)
95% CI	157.2–215.4	168.9–212.3
Min; Max	133.0; 274.8	136.3; 292.5
*p*-value	0.815 ^1^	

^1^ Calculated using an unpaired *t*-test.

**Table 3 ijms-26-03353-t003:** Quantitation of 5-EU-labelled RNA recaptured from experimental and control samples, stratified by EV-recipient cell type.

5-EU-Labelled RNA	Experimental Samples		Control Samples	
	EV-Recipient Chondrocytes (*N* = 5)	EV-Recipient Synoviocytes (*N* = 4)	EV-Recipient Chondrocytes (*N* = 1)	EV-Recipient Synoviocytes (*N* = 1)
RNA concentration, ng/µL				
Mean (SD)	43.1 (5.6)	62.8 (14.1)	0.9 (0.0)	1.8 (0.0)
95% CI	36.2–50.0	40.5–85.2	–	–
Min; Max	33.6; 48.4	54.9; 83.9	–	–
*p*-value	0.016 ^1^		–	

^1^ Calculated using a Mann–Whitney U test. 5-EU, 5-ethynyl uridine.

**Table 4 ijms-26-03353-t004:** Differentially expressed miRNAs between EV-recipient chondrocytes and synoviocytes.

miRNA	logFC	*p*-Value	FDR	Significance
eca-miR-27b	−2.6	<0.001	<0.001	Increased in EV-recipient chondrocytes (RNA originated from EV-donor synoviocytes)
eca-miR-23b	−1.7	0.002	0.033	Increased in EV-recipient chondrocytes (RNA originated from EV-donor synoviocytes)
eca-miR-143	2.6	0.002	0.033	Increased in EV-recipient synoviocytes RNA originated from EV-donor chondrocytes)
eca-miR-31	−4.9	0.003	0.033	Increased in EV-recipient chondrocytes (RNA originated from EV-donor synoviocytes)
eca-miR-21	0.8	0.013	0.110	Increased in EV-recipient synoviocytes (RNA originated from EV-donor chondrocytes)
eca-miR-181a	2.2	0.015	0.110	Increased in EV-recipient synoviocytes (RNA originated from EV-donor chondrocytes)
eca-miR-191a	−0.6	0.016	0.110	Increased in EV-recipient chondrocytes (RNA originated from EV-donor synoviocytes)
eca-miR-181b	1.8	0.018	0.110	Increased in EV-recipient synoviocyte (RNA originated from EV-donor chondrocytes)
eca-miR-199a-5p	−0.6	0.027	0.145	Increased in EV-recipient chondrocytes (RNA originated from EV-donor synoviocytes)

FDR, false discovery rate; logFC, log fold change.

## Data Availability

Data have been submitted to NCBI GEO (GSE292875).

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
