# Peer review of "Extracellular Vesicle-Derived microRNA Crosstalk Between Equine Chondrocytes and Synoviocytes—An In Vitro Approach"

_ijms, 2025, doi:10.3390/ijms26073353_

Round 1
Reviewer 1 Report
Comments and Suggestions for Authors
This is an interesting and relevant study detailing the methods to analyze microRNA crosstalk between chondrocytes and synoviocytes. This is a well-written manuscript with significant contributions to the field. Minor suggestions could improve the readability and flow, which are detailed below:
General comments: Please use passive voice throughout.
- Abstract. Change "we describe" to "this study describes"
- Introduction is well written and detailed the role and function of chondrocytes and synoviocytes, as well as importance of microRNA and EV-mediated exchanges between these cell types. No suggestions for improvement.
- Results are well written. Table 2 could have an accompanying figure that provides an example to demonstrate the size difference between EVs derived from the two cell types but is not needed if not available.
- Results. Figure 2 and Table 4 provide clear and well-defined outcomes and are well described, and Figure 3 demonstrates how well this type of study can delineate specific targets and pathways that may be mediated with this type of cross talk.
- Discussion. Please retain the flow and remain in the passive voice throughout. Line 240 change "we reported" to "this study reported"; line 276 change "we quantified" to "this study quantified"; Line 285 change "in our study" to "this study"; Line 287 change "we were" to "this study was"; Line 289 change "our study" to "this study"; Line 300 change "we found" to "this study found"; Line 316 change "we found" to "this study found"; Line 349 change" our results" to "these results"; Line 352 change "our results" to "these results"; Line 359 change "in our study" to "in this study"; Line 379 change "in our study, we found" to "in this study, it was observed"; Line 383 change "we hypothesize" to "the hypothesis"; Line 388 change "we also found" to "This study also found".
- Discussion. Suggest paragraph break at line 372. New paragraph could begin with..."While the synovium...". Similarly, a new paragraph could begin with line 379. Following the suggested change above a new paragraph would begin with "In this study, it was observed that...".
- Methods. Well described and details allow for potential replication of study results. Well done. Minor issue with g, which should be italicized as g.
- Methods. Is there another study that used the tool "Grow - Diseases and Functions". Perhaps one of the already cited references used this. It would be nice to include a reference or citation to a recently published work that used this methodology.
Overall, this is a sound study with interesting and potentially wide-reaching impact. Very well done.
Reviewer 2 Report
Comments and Suggestions for Authors
General evaluation and characteristics of the reviewed scientific article:
The article addresses the important and timely topic of intercellular communication in the joint via extracellular vesicles (EVs) and the role of microRNAs in the regulation of joint homeostasis and pathological processes. The research methodology is interesting, and the results bring new information to the field of osteoarthrosis research. Despite these merits, the text contains several important shortcomings that should be corrected before publication. Below are my detailed remarks and comments on the article.
Minor comments:
- The introduction is far too short needs to be supplemented with recent publications related to articular cartilage, please expand the first paragraph of the introduction and add current references:
DOI 10.1016/j.ijnonlinmec.2022.104275
DOI 10.3390/healthcare12161648
- The authors did not fully characterize extracellular vesicles in terms of their membrane surface markers. Future work should analyze EV surface markers (e.g., CD9, CD63, CD81) using techniques such as western blot or flow cytometry to confirm their nature. Please describe this limitation and provide a proposal to address it in future work.
- The article provides evidence for the presence of miRNAs in EV and their transfer between cells, but does not analyze the functional effects of this process. In the future, it would be worthwhile to conduct studies on the expression of target genes or proteins in recipient cells after incubation with EV, such as by RT-qPCR or western blot, to confirm the biological activity of transferred miRNAs. I recommend extending the discussion and considering future studies that include this aspect.
- The study was conducted on a limited number of samples (n=8 for chondrocytes, n=9 for synoviocytes), which may affect the statistical power of the analyses. Consideration should be given to increasing the number of samples or at least conducting a statistical test power analysis to assess the reliability of the results obtained.
- The authors used Ingenuity Pathway Analysis (IPA) software, but did not provide detailed filtering criteria and analysis parameters. The miRNA filtering criteria, thresholds, and how signaling pathways are selected in bioinformatics analysis should be further clarified, which would allow better replicability of the results.
- The article focuses only on the horse model, but does not present a comparison with data from studies on human chondrocytes and synoviocytes. In my opinion, it would be worthwhile to add a discussion section, which would discuss analogous studies conducted in humans and the potential translational implications of the results obtained.
- The authors suggest that transfected miRNAs have important effects on inflammatory processes and joint homeostasis, but do not provide functional evidence that changes in miRNA expression lead to specific biological effects. Please consider formulating your conclusions more cautiously and highlighting the limitations of the lack of functional studies.
I congratulate the authors on an interesting paper and wish them continued success.
